# An Oxford Nanopore Technology-Based Hepatitis B Virus Sequencing Protocol Suitable for Genomic Surveillance Within Clinical Diagnostic Settings

**DOI:** 10.3390/ijms252111702

**Published:** 2024-10-31

**Authors:** Derek Tshiabuila, Wonderful Choga, James E. San, Tongai Maponga, Gert Van Zyl, Jennifer Giandhari, Sureshnee Pillay, Wolfgang Preiser, Yeshnee Naidoo, Cheryl Baxter, Darren P. Martin, Tulio de Oliveira

**Affiliations:** 1Centre for Epidemic Response and Innovation (CERI), Stellenbosch University, Stellenbosch 7600, South Africa; wchoga@sun.ac.za (W.C.); naidooy@sun.ac.za (Y.N.); cbaxter@sun.ac.za (C.B.); tulio@sun.ac.za (T.d.O.); 2Duke Human Vaccine Institute, Duke University Medical Center, Durham, NC 27710, USA; sanemmanueljames@gmail.com; 3Division of Medical Virology, National Health Laboratory Service Tygerberg, Faculty of Medicine and Health Sciences, Stellenbosch University, Cape Town 8000, South Africa; tongai@sun.ac.za (T.M.); guvz@sun.ac.za (G.V.Z.);; 4KwaZulu Natal Research and Innovation Sequencing Platform (KRISP), University of KwaZulu Natal, Durban 4001, South Africa; giandharij@ukzn.ac.za (J.G.); pillays11@ukzn.ac.za (S.P.); 5Computational Biology Division, Department of Integrative Biomedical Sciences, Institute of Infectious Diseases and Molecular Medicine, University of Cape Town, Cape Town 8000, South Africa; darrenpatrickmartin@gmail.com

**Keywords:** HBV, recombination, whole-genome sequencing, NGS, ONT

## Abstract

Chronic Hepatitis B Virus (HBV) infection remains a significant public health concern, particularly in Africa, where the burden is substantial. HBV is an enveloped virus, classified into ten phylogenetically distinct genotypes (A–J). Tests to determine HBV genotypes are based on full-genome sequencing or reverse hybridization. In practice, both approaches have limitations. Whereas diagnostic sequencing, generally using the Sanger approach, tends to focus only on the S-gene and yields little or no information on intra-patient HBV genetic diversity, reverse hybridization detects only known genotype-specific mutations. To resolve these limitations, we developed an Oxford Nanopore Technology (ONT)-based HBV diagnostic sequencing protocol suitable for clinical virology that yields both complete genome sequences and extensive intra-patient HBV diversity data. Specifically, the protocol involves tiling-based PCR amplification of HBV sequences, library preparation using the ONT Rapid Barcoding Kit (Oxford nanopore Technologies, Oxford, OX4 4DQ, UK), ONT GridION sequencing, genotyping using genome detective software v1.132/1.133, a recombination analysis using jpHMM (26 October 2011 version) and RDP5.61 software, and drug resistance profiling using Geno2pheno v2.0 software. We prove the utility of our protocol by efficiently generating and characterizing high-quality near full-length HBV genomes from 148 residual diagnostic samples from HBV-infected patients in the Western Cape province of South Africa, providing valuable insights into the genetic diversity and epidemiology of HBV in this region of the world.

## 1. Introduction

The hepatitis B virus (HBV) infection is a significant global health concern, linked to severe outcomes such as hepatocellular carcinoma (HCC) and liver cirrhosis. It is particularly prevalent in sub-Saharan Africa, where approximately 6.1% of adults are chronically infected, contributing to a substantial portion of the 296 million global cases [1]. A recent study, such as that by Ghenea et al. (2021) [2], demonstrated the effectiveness of antiviral therapy in chronic hepatitis B (CHB). This study showed that parameters such as the HOMA index, fasting insulin levels, and baseline HBV DNA are predictive of early viral response in patients undergoing Peg-IFN α-2a/b treatment. Approximately 1.5 million new chronic HBV (CHB) infections occur annually across Africa, accounting for a quarter of new cases globally. In 2019, the global CHB prevalence was estimated at 4.1%, with the Western Pacific region having the highest prevalence (7.1%) and the European region the lowest (1.1%) [1,3].

HBV, a member of the *Hepadnaviridae* family, possesses a compact circular genome of approximately 3.2 kilobases (kb) [4]. It comprises four genes—HBx (X), Core (Pre-C/C), Surface (S), and Polymerase (P) encoded in seven overlapping reading frames. The S protein, containing the antigenicity, or “a”, determinant region within the major hydrophilic region (MHR) is crucial, as it is the primary target of neutralizing HBV antibodies with substitutions in and around this region being associated with immune escape and vaccine failure [5].

To date, ten HBV genotypes (A–J) with an intergroup nucleotide divergence of at least 8% at the whole genome level have been identified [6]. The most prevalent genotypes globally are C (26%), D (22%), E (18%), A (17%), and B (14%) [3]. In Africa, the predominant circulating genotypes include A, D, and E, with sub-genotypes A1 and D3 predominating in southern Africa. Specifically, sub-genotype A1 remains most prevalent in South African populations (accounting for 97% of infections) and has been linked to severe liver disease and rapid progression to HCC [7,8].

While whole-genome sequencing-based surveillance is becoming a key tool for understanding the distribution, prevalence, and evolution of viral pathogens [9,10,11], it has yet to be fully leveraged in clinical diagnostic settings. In many settings, rapid point-of-care screening tests are used to detect the hepatitis B surface antigen (HBsAg) in serum or plasma as a marker of active infection. However, a major concern when using diagnostic tests is that they must possess a high degree of sensitivity and an acceptable level of specificity to reduce false results [12,13]. For more detailed characterization of HBV in patient samples, Sanger sequencing of complete or partial HBV genomes is considered the gold standard and has been used to classify HBV into its ten genotypes. However, it is, often restricted to analyzing specific genes and is rarely used for the analysis of intra-patient genetic diversity [14]. However, partial genome sequences can be somewhat misleading when characterizing recombinant HBV genomes, and Sanger sequencing yields little or no information on intra-patient HBV genetic diversity: information which would be extremely valuable with respect to monitoring the emergence of drug resistance or immune evasion mutations, estimating the durations of chronic infections, understanding the progression of pathogenesis, and tracing transmission patterns [14]. Illumina deep sequencing, while effective for genotyping and characterizing genetic diversity [15], suffers from limitations such as the inability to sequence long DNA stretches, biases introduced during amplification steps, and challenges in generating sufficient overlap between DNA fragments [16].

High-throughput sequencing (HTS) techniques are powerful tools that, in addition to diagnosing CHB infections and genotyping HBV, would both enable the accurate characterization of recombinant HBV genomes (including those with mixed genotypes) and provide detailed data on intra-patient HBV genetic diversity [17]. Among numerous other HBV-focused applications, HTS and downstream analyses have previously been used to sequence complete HBV genomes [9,11,18,19], track the demographics of HBV populations within individual CHB patients [20], and identify the prevalence of drug-resistance mutations in large patient cohorts [14,21,22,23].

Two major challenges associated with HTS workflows employed to generate viral genome sequence data are (1) the efficient and accurate barcoding of samples needed for multiplexed sequencing where multiple patient samples are simultaneously sequenced in a single run and (2) the accurate reassembly of sub-genome-length sequence reads into complete genomes. Third-generation HTS technologies, such as the Oxford Nanopore Technology (ONT) MinION, GridION, and PromethION, have largely overcome these limitations. Barcoding kits allow for cost-effective and efficient sequencing by enabling the pooling and running of multiple libraries on a single flow cell. Different types of barcoding kits are available, including ligation-based, PCR-based, and rapid chemistry-based kits, each with its own advantages and input requirements (https://community.nanoporetech.com/docs (accessed on 20 November 2023)). A study by McNaughton et al. describes advancements in a sequencing protocol utilizing isothermal rolling-circle amplification and ligation-based barcoding kits [24]. The ONT rapid barcoding kit does not require individual sample washes and allows samples to be processed uniformly without quantification or normalization [25]. The sample runtime on ONT, per 96 samples, is almost half that of Illumina (14 h compared to 26 h), mainly due to a real-time data analysis with ONT. The rapid barcoding library preparation utilized by ONT also requires fewer reagents, as everything is contained within the kit, and is thus cheaper than Illumina sequencing [26]. Further, the sequence reads generated using ONT are substantially longer than those generated by Illumina, a factor that vastly simplifies the assembly of whole genome sequences. However, ONT still exhibits a higher error rate than Illumina sequencing, although this has improved with newer chemistry and the use of post-sequencing software such as Nanopolish [27].

Here, we describe an optimized the ONT-based HBV whole genome sequencing protocol, and associated downstream computational analyses that will be applicable within a clinical HBV diagnostic setting. Using 148 samples from chronic South African CHB patients we demonstrate that the protocol enables accurate recombination-aware genotyping of patient samples and the detection of drug-resistance mutations.

## 2. Results

### 2.1. Sequencing of Samples

To evaluate the applicability of our ONT-based protocol in a clinical setting, we sequenced 148 HBV-positive samples collected between 19 September 2022 and 5 November 2023 in Western Cape, South Africa. The average age of the patient cohort was 40 years (range: 0–69), with a gender distribution of 37.2% females and 60.8% males, and three samples had no gender information. The overall median viral load (VL) was 16,368 IU/mL (IQR: 1452–1,445,182). Of the 148 samples sequenced, 146 HBV genomes (98.6%) were obtained. Eight samples with missing age, gender, or VL information were excluded from the analyses, and the remaining 138 sequences with available metadata were used for subsequent analyses (Table 1).

The 138 near-complete HBV genomes with available metadata had a median sequencing depth of 2344 (IQR 584–13,497) and median genome coverage of 98.33% (IQR 92–100%). Of the 138 sequences, 124 (89.86%) had a uniform coverage of greater than 80%. The Genome Detective (GD) Hepatitis B phylogenetic typing tool classified 114 (82.60%) of the 138 near-complete genomes and determined that the majority (n = 103, 90.40%) were genotype “A”, followed by “D” (n = 9, 7.89%) and “E” (n = 2, 1.75%). The 24 genomes (17.39%) that GD could not assign to a genotype level had a degree of genome coverage that was significantly lower than that of the genomes that could be assigned (median coverage of 92.74% vs. 98.68%; *p* = 0.003472; Wilcoxon rank sum test). Further analyses of genome coverage revealed a significant difference in coverage between genotypes A and D (*p* = 0.00884) (Figure 1A). Although VL did not significantly affect the genome coverage, a weak linear association (R^2^ = 0.099; Pearson linear correlation) was observed between high VL and genome coverage (Figure 1B).

### 2.2. Recombination Detection and Breakpoint Identification

The recombination detection program (RDP) version 5.46 and the Jumping Profile Hidden Markov Model (jpHMM) HBV tool (26 October 2011 version) were used to identify recombinant sequences and infer breakpoints. jpHMM maps were generated for the 24 “unassigned” study sequences to infer evidence of inter-genotype recombination. Of the 24 sequences, traces of recombination were detected in 16, and the remaining eight sequences were assigned to genotypes by jpHMM (6 D, 1 A, 1 G) (Figure 2). jpHMM classified twelve sequences as A/D recombinants (i.e., the parents of the recombinants belonged to genotypes A and D) (Figure 3), three as A/E recombinants, and one as an A/D/G recombinant (Figure 4A). Bootscan plots (Figure 4B) and genome coverage maps (Figure 4C) were obtained from the GD Hepatitis B phylogenetic typing tool to assess the accuracy of the non-A/D recombination classifications.

Further assessment of recombination using RDP5.46 but without assuming only recombination between subtypes (i.e., also accounting for intra-genotype recombination) confirmed that 12 of the 16 recombinants identified by jpHMM were actual recombinants (Figure 2). *p*-values are listed in the Appendix A, Table A1. Of the twelve sequences classified as A/D recombinants by jpHMM, RDP5.46 confirmed seven as A/D recombinants, with the remaining five classified as two A/D/A recombinants, one an A/A recombinant, and two as non-recombinants. Two A/E recombinants were classified as A/D and A/D/A recombinants by RDP5.46 (Figure 2). The recombination region count matrix produced by RDP5.46 illustrated heightened susceptibility for recombinational transfers of specific genomic regions, particularly the end of the pol region, the X region, the pre-C region, and the C region (Figure 5A). Recombination events were distributed variably, with recombination breakpoints concentrated towards the end of the P gene, within the X and pre-C regions, and at the start of the C region (Figure 5B). 

### 2.3. Drug Resistance and Immune-Evasion Profiling

Mutations related to HBsAg escape and drug resistance in the overlapping RT/HBsAg genome region were determined using Geno2pheno-HBV. Among the 138 sequences, thirteen (9.42%) were likely resistant to lamivudine and telbivudine. Ten (7.25%) were resistant to entecavir, and nine (6.52%) were resistant to adefovir. However, all sequences were likely susceptible to tenofovir (Figure 6A). 204V and 180M are the most prevalent drug-resistance mutations (Table 2). The most common vaccine escape HBsAg mutation is 100C, which is responsible for HBV detection failure [29], with a prevalence of five (3.62%). The second most prevalent HBsAg mutation is 120T, which is responsible for vaccine, immunotherapy, and diagnostic detection failure, with a prevalence of two (1.45%). This is followed by mutations 128V, 109I, and 101H, each with a prevalence of one (0.72%), which are responsible for vaccine escape (Figure 6B) [29].

## 3. Discussion

In this study, we demonstrate that ONT sequencing, utilizing the Oxford Nanopore Rapid Barcode Kit, enables the rapid and simple generation of full-length HBV genomes. Additionally, we illustrate the utility of the rich sequencing data generated by this approach in the recombination-aware genotyping of HBV genomes and the detection of mutations associated with drug resistance and vaccine/immunotherapeutic escape.

Our protocol yielded 124 genomes with uniform coverage of greater than 80% and a sequencing depth of approximately 2343.86. We opted for ONT sequencing, which overcomes Sanger and Illumina limitations by providing long-read sequencing, eliminating the need for complex library preparation processes, and reducing the risk of biases associated with amplification steps. As the ONT rapid barcoding library preparation requires fewer reagents than the Illumina sequencing protocol, ONT sequencing was also much cheaper (Illumina cost per sample is ~150–250 USD while the ONT cost per sample is ~10–40 USD) [26].

Full-length genome and sub-genomic ONT-based sequencing approaches have previously been used for HBV [24,30,31]. In these studies, ONT sequencing only worked well for samples with high HBV loads, with one study [31] having a raw read error rate of ~12% and another unable to definitively confirm putative minority variants detected in the MinION reads [30,31]. In addition to ONT-based approaches, Dopico et al. (2021) developed an HBV sequencing protocol using the Distance-Based discrimination method (DB Rule) on Illumina MiSeq to sequence-specific, relatively short regions of the HBV genome (preS and 5′end of the HBV X gene regions), rather than generating complete genomes [32]. This method is valuable for the rapid identification of variants in targeted regions but does not offer the comprehensive genomic insights that whole-genome sequencing approaches enable. An important factor associated with the successful sequencing of viral genomes is the VL present in samples: the higher the VL in a sample, the higher the yield of amplified products to be sequenced and the easier the assembly of a complete genome [33]. One of the studies [24] developed a sequencing protocol utilizing ONT ligation-based barcoding kits that improved the accuracy of HBV nanopore sequencing for use in research and clinical applications. Our rapid, chemistry-based, barcoding kit, sequencing protocol produced complete genomes at low (<2000 IU/mL), medium (2000–20,000 IU/mL), and high (>20,000 IU/mL) VL and allowed for the identification of various HBV genotypes, including genotypes A, D, and E.

Genome characterization of a virus can be important for clinical diagnostics as, beyond identifying the infecting agent, it can reveal clinically relevant genetic variations [34]. HBV-A was the most prevalent genotype among our study cohort, which is consistent with this genotype’s high prevalence in sub-Saharan Africa [35]. A study by Jose-Abrego et al. (2021) explored the possible influence of HIV-HBV co-infection, which is likely to be prevalent in South Africa, on HBV genomes [36]. They observed that co-infection could lead to genotype mixtures, increased viral load, and more severe liver damage, suggesting that HIV coinfection may have important implications for HBV genome diversity and clinical outcomes. A recombination analysis also revealed complex viral replication/recombination dynamics, with 16 identified recombinants, primarily between genotypes A and D (A/D). This was to be expected due to the high prevalence of genotypes A and D in Southern Africa [7,8]. While the RDP5.46 analysis confirmed eight of these recombinants, discrepancies were noted, highlighting the challenges in precisely classifying recombinant strains. Although recombination can be easily detected using phylogenetic trees and recombination software, it is much more difficult to determine whether detected recombinants exist or whether they are detection artifacts arising from (1) primer jumping during Polymerase Chain Reaction (PCR) of samples that are either from mixed infections or have been accidentally cross-contaminated, (2) “backfilling” of failed amplicons with contaminating sequence reads or (3) incorrectly assembled genomes where reads from multiple different genetically distinct viruses get assembled into a single genome. In general, the only way to confirm the existence of a recombinant is to independently amplify and re-sequence samples or to detect multiple genomes of the same recombinant lineage in different patients (i.e., by identifying circulating recombinants). Of the 16 recombinants detected, only three were identified by RDP5.46 as circulating recombinants. Therefore, until the other 13 are independently amplified and sequenced, it cannot be definitively stated that they are not simply either sequence amplification or sequence assembly artifacts.

Nevertheless, the fact that HBV recombinants have been so widely and frequently detected suggests that the recombinants detected in this study are likely real. Genotype B/E recombinants identified in Nigeria and Eritrea are the most common in Africa [37]. Genotype D/E recombinants are also common, having been identified in eight countries, namely Kenya, Niger, Egypt, Ghana, Libya, Mali, Eretria, and Uganda [37], followed by genotype A/E, identified in five countries, namely Uganda, Eretria, Ghana, Niger, and Mozambique [37,38], and genotype A/D, reported in Egypt, Eretria, and Uganda [37].

Others have also detected recombination hotspots within the C region, pre-C, P, and X genes [39]. These regions were also frequently transferred during recombination events and, as a result, may impact the diagnosis and treatment of HBV since coinfection and viral recombination can trigger greater virulence and result in a worsened patient clinical status [33].

The Geno2pheno-HBV tool for identifying HBsAg vaccine escape and drug-resistant mutations [40,41] identified various potential drug resistance, HBsAg vaccine/immunotherapeutic escape, and diagnostic failure-associated mutations in our study population. Mutations, such as the triple mutation 173L + 180M + 204I/V, and 133L/T, found in the Pol gene, have been identified as major vaccine escape mutations. An HBV study in Bangladesh identified HBsAg mutant 128V as their most common mutant [29], while mutant 100C was the most prevalent in our study. This may be due to the difference in genotype prevalence between the two study populations, as HBV-C was the most prevalent genotype in Bangladesh, whilst genotype A is the most prevalent among our study population. One of the most prevalent resistance-associated mutations for HBV is 204V/I for both treatment-experienced and treatment-naive individuals. This mutation can either occur alone or can occur in combination with other mutations such as 80I/V, 173L, 180M, 181S, 184S, 200V, and 202S [42]. Our study also notes a high prevalence of 204V/I in combination with other drug-resistance mutations.

First-line treatment for CHB includes PEGylated interferon and nucleoside/nucleotide analogs such as Tenofovir, Entecavir, and Lamivudine [8]. In South Africa, over 1.9 million people are chronically infected with HBV, and the most used first-line treatment is Tenofovir in the form of tenofovir disoproxil fumarate (TDF) [8]. In this study, we note that all sequences suggested susceptibility to Tenofovir. TDF treatment is long-term and is frequently given indefinitely due to the risk of infections reactivating when therapy is terminated, as such treatments do not entirely remove the replication-competent viral genomes [8,43]. A CHB functional cure, characterized by loss of HBsAg and reduced risk of HCC, can be achieved by treatment regimens including TDF [8]. Resistance to TDF has been noted in patients harboring mutations; 80M, 180M, 204V/I, 200V, 221Y, 223A, 184A/L, 153Q, and 191I [44]. The absence of drug-resistance mutations in genotype E genomes and their presence in genotype A and D genomes highlight the importance of monitoring drug resistance, as treatment with TDF may not be successful for these patients.

A major limitation of the current study is that HBV samples were only sequenced using the proposed protocol. The sequencing data were not compared to results generated using Illumina sequencing protocols. Moreover, efforts have been invested in optimizing the analysis pipeline to enhance accessibility and enable clinical laboratory staff to execute the entire process from start to end seamlessly. This initiative seeks to empower non-specialist bioinformaticians and streamline workflows, making genomic analysis more user-friendly and ensuring that valuable insights can be obtained without a dependency on specialized expertise. The aim is to make the application of these advanced sequencing technologies accessible to all, allowing broader adoption and application in clinical settings.

While short-read sequencing is the most used form of NGS [28], despite ONT having a slightly higher error rate, ONT appears to generate high-quality data at a very affordable cost. Therefore, ONT-based sequencing is presently the most cost-effective HTS technology, especially well-suited for countries with limited resources for monitoring shifting viral demographics and tracking the prevalence and spread of drug resistance and vaccine evasion mutations.

The high value of genome surveillance was excellently demonstrated during the COVID-19 pandemic. Although whole genome sequencing was primarily used to monitor virus evolution, the prospect of agnostically diagnosing any viral pathogen (whether known or unknown) through HTS-based whole genome sequencing is likely to be realized within the coming decade. By addressing the limitations of conventional HBV diagnostic methods, our ONT-based protocol, which efficiently generates high-quality near full-length genomes, offers a robust solution for both full-genome sequencing and in-depth analyses of intra-patient genetic diversity, something previously limited in clinical practice. This approach is not only critical for improving HBV diagnosis and treatment strategies but also serves as a model for how NGS technologies can be integrated into routine viral surveillance. Such advancements in viral pathogen diagnostics are vital as we move toward a future where high-throughput sequencing could routinely enable comprehensive, agnostic detection of infectious diseases in clinical settings.

## 4. Materials and Methods

### 4.1. Study Design

The 148 anonymized residual diagnostic plasma/serum samples analyzed in this cross-sectional study were obtained from the National Health Laboratory Service (NHLS) based at the Division of Medical Virology, Tygerberg Hospital, South Africa. These samples had been sent for HBV DNA load measurement, and confirmed to be HBV DNA-positive. The study was approved by the Stellenbosch University Health Research Ethics Committee (HREC: N22/08/089).

### 4.2. HBV DNA Extraction

DNA was extracted from 1 mL of patient serum using the Qiagen DNA extraction kit, following the manufacturer’s instructions (QIAGEN, Hilden, Germany). Extracted DNA was eluted in a volume of 50 μL. The extracts were stored at −80 °C until use.

### 4.3. Primer Design

Primers were designed using Primal scheme (https://primalscheme.com (accessed on 20 June 2023)) using the HBV sequences of genotypes A–J obtained from GenBank (https://www.ncbi.nlm.nih.gov/labs/virus/vssi/#/ (accessed on 20 June 2023)). These primers were designed to target the full length of the HBV genome (Table 3) and were used to obtain tiling PCR products for ONT library preparation.

### 4.4. Tiling-Based Polymerase Chain Reaction

For HBV whole-genome amplification using a multiplex PCR approach, we designed primers that would generate 1200 base pair (bp) amplicons with 70 bp overlaps, spanning the 3200 HBV genome. Since HBV has a relaxed circular DNA genome, we modified the ARTIC SARS-CoV-2 amplification protocol with proven applicability in clinical settings [45] by excluding the synthesis of complementary DNA (cDNA). Instead, we prepared master mixes using two pools of HBV primers. The ARTIC protocol utilizes these primer pools to amplify tiled amplicons of approximately 400 bp [46]. Randomly generated PCR products were quantified using the Qubit double-strand DNA (dsDNA) High Sensitivity assay kit on a Qubit 4.0 instrument (Life Technologies Corporation, Chicago, IL 60693, USA). Amplicons were purified using 1× AMPure XP beads from Beckman Coulter and quantified using the Qubit dsDNA HS assay kit from ThermoFisher (Waltham, MA 02451, USA). The DNA library preparation was carried out using the SQK-RBK110.96 ligation sequencing kit from ONT (Oxford, OX4 4DQ, UK), and rapid barcoding using SLK RBK109. The resulting sequencing libraries were loaded into an R9.4 flow cell from ONT.

### 4.5. Raw-Read Assessment and Genotyping

A whole genome sequencing of samples was performed using the GridION platform (ONT, Oxford, UK), using the primers listed in Table 3, and the samples were sequenced in three runs. Subsequently, raw sequencing files underwent base calling using Guppy v3.4.5, and barcode demultiplexing was performed using qcat (demultiplexing tool) incorporated in MinKNOW Release (version 22.05.12). de novo genome assembly was carried out using GD v1.132/1.133 (https://www.genomedetective.com/ accessed on 3 August 2023). Briefly, GD utilizes DIAMOND to identify and classify potential viral reads within broad taxonomic units. The viral subset of the SwissProt UniRef protein database was used for this classification. Subsequently, reads were assigned to candidate reference sequences using the National Center for Biotechnology Information (NCBI) nucleotide database to implement a Basic Local Alignment Search Tool analyses (BLAST) and aligned using AGA (Annotated Genome Aligner) and MAFFT (v7.490) [47]. The resulting contigs and consensus sequences were then exported in FASTA file format. Raw read files were deposited in the NCBI SRA database (SRR26038114-SRR26038214, SRR2746800-SRR2746843).

### 4.6. Recombination Analysis

All sequences in the study were assessed for inter-genotype recombination using the jpHMM-HBV online tool, available at http://jphmm.gobics.de/submission_hbv (accessed on 23 November 2023), and intra- and inter-genotype recombination using RDP5.46 [48]. The jpHMM-HBV online tool was used to plot and visualize inter-genotype recombination breakpoints. A full exploratory automated scan for recombination with RDP5.46 was performed with the RDP [49], GENECONV [50], and MaxChi [51] methods as “primary scanning methods” to detect recombination signals and the Bootscan [52], Chimaera [53], SiScan [54], and 3Seq [55] methods to verify the signals (secondary scanning methods). The RDP5.46 general setting for sequence type was set as circular, the bootscan window size was set to 500 bp with a step size of 20 bp, and the SiScan window size was set to 200 bp with a step size of 20 bp. The remaining analysis options were kept at their default values. To ensure reliability, the HBV sequences identified as potential recombinants by RDP5.46 were only considered to be recombinant when the recombination signal was supported by at least four recombination detection methods with *p*-values of ≤0.05 after Bonferroni correction for multiple comparisons [56,57,58].

### 4.7. Genotyping and the Identification of Drug-Resistance and Immune-Escape Mutations

The GD HBV phylogenetic typing tool, available at https://www.genomedetective.com/app/typingtool/hbv/ (accessed on 18 November 2023), was used to infer genotypes, sub-genotypes, and serotypes of HBV genomes and Geno2pheno software version 2.0, available at http://hbv.geno2pheno.org/ (accessed on 18 November 2023), was used to identify potential drug resistance and antibody escape mutations. Briefly, the HBV phylogenetic typing tool utilizes phylogenetic methods to identify the HBV genotype of a nucleotide sequence. Geno2pheno is an online platform that applies an algorithm that is widely used to predict phenotypic drug resistance mutations in the P open reading frames and antibody escape mutations in the S open reading frames of HBV genomes. The sequences from this study were deposited in GenBank under the accession numbers PP123755–PP123892.

### 4.8. Statistical Analyses

All numerical data were analyzed using RStudio version 4.1.3 (Posit team (2023). RStudio: Integrated Development Environment for R. Posit Software version 4.1.3, PBC, Boston, MA. URL http://www.posit.co/ (accessed on 10 October 2024)). Baseline characteristics were presented as proportions for categorical data, as means for normally distributed continuous data, or as medians for skewed normally distributed variables. Kruskal–Wallis tests (for skewed continuous variables) were used to test for differences in age and viral load groupings. A Chi-square test was used to determine the significant difference in the gender classes. A Pearson linear correlation test was used to test for a correlation between viral loads and genome coverage of viral sequences and a pairwise *t*-test was used to test for differences in genome coverage between the different HBV genotypes.

## Figures and Tables

**Figure 1 ijms-25-11702-f001:**
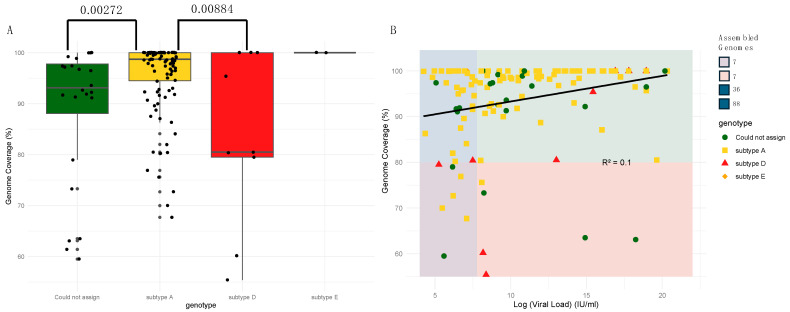
(**A**) Comparison of genome coverage for different detected HBV genotypes. The boxes indicate the lower quartile, median, and upper quartile minimum and maximum values by the whiskers. Significant differences between genome coverage (paired Wilcoxon test) values between each genotype are denoted above the box and whisker plots. (**B**) Scatter plot for log viral load against the genome coverage for detected HBV genotypes. A total of 124 genomes with >80% coverage was produced (36 with a log viral load of <7.6 IU/mL and 88 with a log viral load of >7.6 IU/mL). HBV genotypes are represented by different colors and shapes. The different background color shades represent different quality control groups. The blue and purple shades represent ≤7.6 IU/mL log viral load and coverage of ≥80% and <80%, respectively. The green and pink shades represent a log viral load of >7.6 IU/mL and coverage of ≥80% and <80%, respectively.

**Figure 2 ijms-25-11702-f002:**
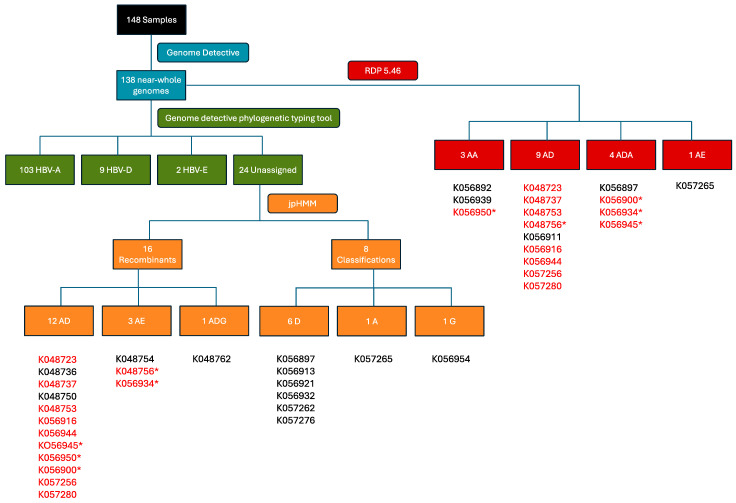
Flowchart showing the identification of recombinants using the jpHMM HBV tool and RDP5.46. The blue section shows the number of genomes produced by the genome detective, the green section shows genotype variation as assessed by the genome detective hepatitis B phylogenetic typing tool; the orange section highlights the jpHMM recombination classifications of the unclassified sequences, and the red section shows the RDP5.46 recombination analysis. Sample IDs are shown below the flowchart sections, and IDs that are shown in red highlight recombinants that both jpHMM and RDP5.46 detected. Sample IDs marked with a “*” were classified as different recombinants by jpHMM and RDP5.46.

**Figure 3 ijms-25-11702-f003:**
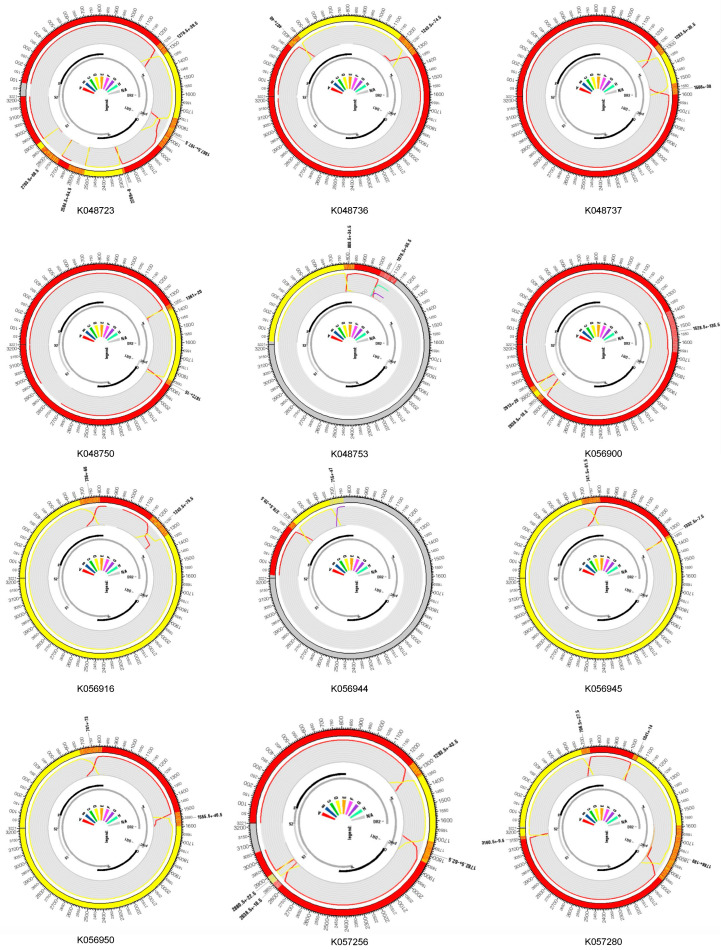
jpHMM genome maps for A/D recombinant viruses. The query isolates identifier names which are listed below each jumping profile map. Genome maps presented here were created using the software package, Circos [28]. The colored shadings represent different HBV genotypes (red = A, yellow = D, gray = unknown). Regions of orange shading represent recombination breakpoint intervals, e.g., region 405 ± 40 (outer ring). All sequence position numbers are given relative to the HBV reference genome AM282986. Positions of genes in the genome are marked with gray and black bars (inner ring). The color legend is in the middle, and the “NA” denotes “not assigned”.

**Figure 4 ijms-25-11702-f004:**
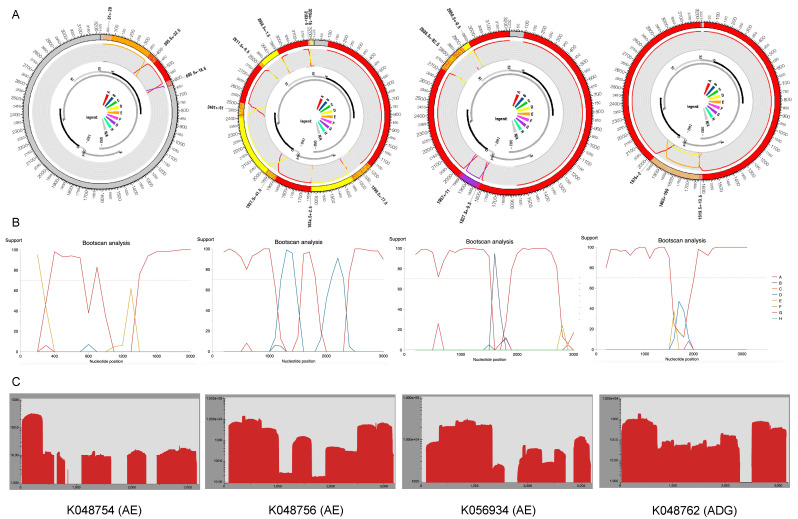
(**A**) jpHMM genome maps and (**B**) genome detective bootscan plots for non-A/D recombinant viruses. Bootscan analysis was performed with a window size of 400 and a step size of 100. (**C**) Genome coverage maps highlighting HBV sequencing depth. The query isolates are listed below each genome coverage map.

**Figure 5 ijms-25-11702-f005:**
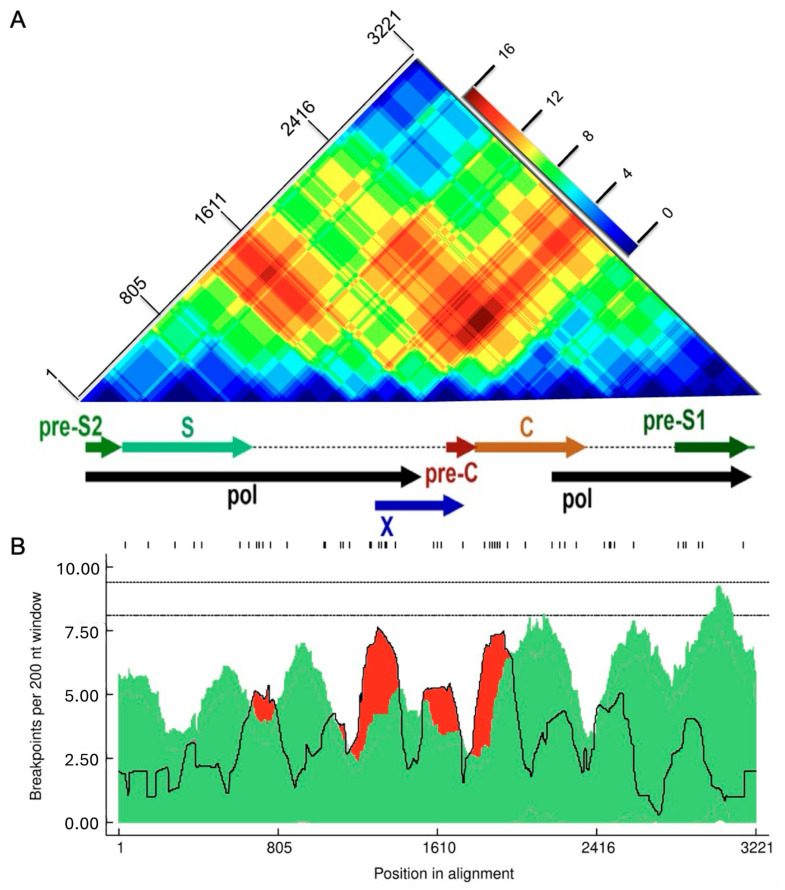
Recombination region and breakpoint distributions (**A**) recombinant region count matrix highlighting areas of the genome that are most and least commonly transferred during recombination events. Unique recombination events were mapped onto a region count matrix based on determined breakpoint positions. Each cell in the matrix represents a pair of genome sites, with the colors of cells indicating the number of times recombination events separated the represented pairs of sites. (**B**) Breakpoint distribution across HBV genomes. All detectable breakpoint positions are represented as black lines above the graph. The green areas show the 99% confidence interval for breakpoint clustering under random recombination. The upper dotted line represents the global 99% confidence interval for a breakpoint clustering under random recombination, and the lower dotted line is the global 95% confidence interval for breakpoint clustering under random recombination. The black line represents the number of breakpoints within a 200-nucleotide window moved along the genome. Areas in red where the black line emerges above the green area are considered recombination warm spots, and those that traverse the dotted global 95% confidence interval line are considered statistically supported recombination hotspots. An HBV gene map is plotted between the figures.

**Figure 6 ijms-25-11702-f006:**
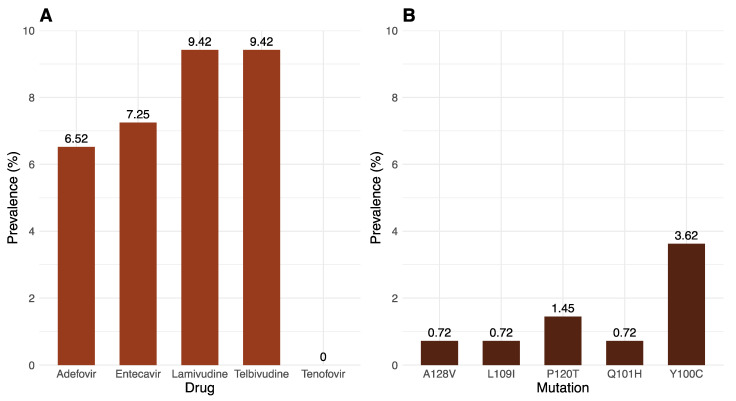
(**A**) Prevalence of predicted drug resistance based on mutation patterns in the RT/HBsAg overlapping region for HBV genomes sampled in South Africa between September 2022 and November 2023. (**B**) Prevalence of HBsAg vaccine escape mutations in the RT/HBsAg overlapping region for these same genomes.

**Table 1 ijms-25-11702-t001:** Demographic characteristics for the 138 patient samples.

Characteristic	Total N = 138 (%)
**Gender**	
Male	86 (62.32)
Female	52 (37.68)
**Age**	
<18	2 (1.45)
18–24	6 (4.35)
25–34	30 (21.74)
35–44	44 (31.88)
≥45	56 (40.58)
**HBV viral load (IU/mL)**	
<2000	39 (28.26)
2000–20,000	31 (22,46)
>20,000	68 (49.28)

**Table 2 ijms-25-11702-t002:** Prevalence of drug-resistant mutations in the RT/HBsAg overlapping region.

Mutation	Adefovir	Entecavir	Lamuvidine	Telbivudine
180M	0	4	4	0
204V	0	5	5	5
181T	1	0	0	0
202K	0	2	0	0
250S	0	1	0	0
250N	0	1	0	0

**Table 3 ijms-25-11702-t003:** The primer sequences and genomic targets used for the tiling PCR and library preparation for ONT sequencing ^1,2^.

Primer Name	Sequence	Genomic Positions (bp)
SC_1_LEFT	TTC CAC CAA GCT CTG CAA GATC	11–32
SC_1_RIGHT	AGAGGAATATGATAAAACGCCGCA	384–407
SC_2_LEFT	CATCATCATCAT CACCA CCTCC	325–346
SC_2_RIGHT	AAAGCCCTACGAACCACTGAAC	692–713
SC_3_LEFT	AAATACCTATGGGAGTGGGCCT	632–653
SC_3_RIGHT	TTGTGTAAATGGAGCGGCAAAG	1655–1676
SC_4_LEFT	AGAAAACTTCCTGTTAACAGACCTATTG	949–976
SC_4_RIGHT	GGACGACAGAATTATCAGTCCCG	1326–1348
SC_5_LEFT	TCCATACTGCGGAACTCCTAGC	1265–1286
SC_5_RIGHT	TGTAAGACCTTGGGCAGGATTTG	1632–1654
SC_6_LEFT	CTTCTCATCTGCCGGTCCGTGT	1559–1580
SC_6_RIGHT	AGAAGTCAGAAGGCAAACGAGA	1947–1970
SC_7_LEFT	GGCTTTGGGGCATGGACATT	1890–1909
SC_7_RIGHT	ATCCACACTCCGAAAGAGACCA	2256–2277
SC_8_LEFT	GACAACTATTGTGGTTTCATATTTCT	2193–2218
SC_8_RIGHT	TTGTTGACACCTATTAATAATGTCCTCA	2576–2594
SC_9_LEFT	TGGGCTTTATTCCTCTACTGTCCC	2492–2515
SC_9_RIGHT	GGGAACAGAAAGATTCGTCCCC	2889–2910
SC_10_LEFT	TTGCGGGTCACCATATTCTTGG	2816–2837
SC_10_RIGHT	GGCCTGAGGATGACTGTCTCTT	3189–3210

^1^ See the following link for details: (https://dx.doi.org/10.17504/protocols.io.5qpvo3xxzv4o/v1 (accessed on 21 June 2023)). ^2^ Primers for Pools 1 and 2. Pool one primers are represented by odd numbers (SC_1, SC_3, …), and pool two are represented by even numbers (SC_2, SC_4, etc.).

## Data Availability

The raw read files generated during this study have been deposited in the NCBI SRA (SRR26038114–SRR26038214, SRR2746800–SRR2746843). The processed sequences have been submitted to GenBank and are accessible under accession numbers PP123755–PP123892.

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
