# Peer review of "An Oxford Nanopore Technology-Based Hepatitis B Virus Sequencing Protocol Suitable for Genomic Surveillance Within Clinical Diagnostic Settings"

_ijms, 2024, doi:10.3390/ijms252111702_

Round 1

Reviewer 1 Report

Comments and Suggestions for Authors

he paper presents a novel protocol for Hepatitis B Virus (HBV) genome sequencing, using Oxford Nanopore Technology (ONT), specifically designed for clinical diagnostics. This protocol aims to overcome limitations in traditional sequencing methods by enabling comprehensive genomic analysis and surveillance within clinical settings. 

Minor grammar issues were observed such as inconsistent usage of “and/or” and slight awkward phrasing in sections like “DNA library preparation.”

The study does not compare the ONT protocol results with other sequencing platforms like Illumina, limiting the assessment of accuracy and reliability.

Please insert at line 45 a phrase about the markers of efectiveness of markers in VHB therapy, like "In recent studies, the role of clinical and biochemical markers in evaluating the effectiveness of antiviral therapy in chronic hepatitis B has been explored. Ghenea et al. (2021) (PMID: 34440963) demonstrated that parameters such as HOMA index, fasting insulin levels, and baseline HBV DNA are predictive of early viral response in patients undergoing Peg-IFN α-2a/b treatment​".

Author Response

Comments 1: Minor grammar issues were observed such as inconsistent usage of “and/or” and slight awkward phrasing in sections like “DNA library preparation.”

Responses 1: We would like to thank the reviewer for the comment and have removed /or found in line 359. We, however, do not see how to edit "DNA library preparation" as that is the process used to prepare the HBV DNA libraries.

Comments 2: The study does not compare the ONT protocol results with other sequencing platforms like Illumina, limiting the assessment of accuracy and reliability.

Responses 2: Thank you for pointing this out, we agree with this comment. It is thus included as one of the limitations found in line 376.

Comments 3: Please insert at line 45 a phrase about the markers of efectiveness of markers in VHB therapy, like "In recent studies, the role of clinical and biochemical markers in evaluating the effectiveness of antiviral therapy in chronic hepatitis B has been explored. Ghenea et al. (2021) (PMID: 34440963) demonstrated that parameters such as HOMA index, fasting insulin levels, and baseline HBV DNA are predictive of early viral response in patients undergoing Peg-IFN α-2a/b treatment​".

Responses 3: We thank you for this comment and have inserted the line and have incorporated the reference.

Reviewer 2 Report

Comments and Suggestions for Authors

In general genotyping of chronic hepatitis B patients is not worthwhile in clinical practice. Nevertheless if there is a correlation with a specific genotype to HCC or impending liver failure in cirrhotic patients early detection could be life saving. Therefore have your group of scientists in South Africa assessed this in your study patients?

Author Response

Comments 1: In general genotyping of chronic hepatitis B patients is not worthwhile in clinical practice. Nevertheless if there is a correlation with a specific genotype to HCC or impending liver failure in cirrhotic patients early detection could be life saving. Therefore have your group of scientists in South Africa assessed this in your study patients?

Responses 1: We thank the reviewer for highlighting this and agree with the comment. We, unfortunately, did not assess this as patient clinical data was limited. This could be a great idea for future studies.

Reviewer 3 Report

Comments and Suggestions for Authors

The introduction of novel methods to the clinical diagnostics is highly important, therefore the manuscript: "An Oxford Nanopore Technology-Based Hepatitis B Virus Sequencing Protocol Suitable for Genomic Surveillance Within Clinical Diagnostic Settings" brings interesting data on implementation of Nanopore sequencing into routine testing. The introduction gives sufficient insight into the subject and the methods are thoroughly described. The sample size is substantial, however, there are a few things with the presentation of the results that caught my attention. I realize that the authors are establishing the protocol for a new test, but it would still be interesting to know a bit more background on the study participants; for example, are they in the clinical care for some time now, treated or are they newly diagnosed? Have any of the participants been previously sequenced (genotyping ore resistance testing using Sanger method) or has the genotype been determined using reverse hybridization?  If any previous data on the participants exist, I believe it would be useful to compare them to the data yield in this research and check for any discrepancies. The comparison shown in Figure 6. does not make to much sense to me if the samples were randomly collected from the patients coming for a routine checkup. If all the patients have been newly diagnosed it would give some insight into the genotypes circulating in the population, but I still find that just showing the percentages of all the genotypes and recombinants in this study group is enough and advise the authors to remove the figure in question. The last part of the discussion, which I assume stands for conclusion should focus more on the HBV. Other than that I find the article acceptable for publication.

Author Response

Comments 1: The introduction of novel methods to the clinical diagnostics is highly important, therefore the manuscript: "An Oxford Nanopore Technology-Based Hepatitis B Virus Sequencing Protocol Suitable for Genomic Surveillance Within Clinical Diagnostic Settings" brings interesting data on implementation of Nanopore sequencing into routine testing. The introduction gives sufficient insight into the subject and the methods are thoroughly described. The sample size is substantial, however, there are a few things with the presentation of the results that caught my attention. I realize that the authors are establishing the protocol for a new test, but it would still be interesting to know a bit more background on the study participants; for example, are they in the clinical care for some time now, treated or are they newly diagnosed? Have any of the participants been previously sequenced (genotyping ore resistance testing using Sanger method) or has the genotype been determined using reverse hybridization?  If any previous data on the participants exist, I believe it would be useful to compare them to the data yield in this research and check for any discrepancies. The comparison shown in Figure 6. does not make to much sense to me if the samples were randomly collected from the patients coming for a routine checkup. If all the patients have been newly diagnosed it would give some insight into the genotypes circulating in the population, but I still find that just showing the percentages of all the genotypes and recombinants in this study group is enough and advise the authors to remove the figure in question. The last part of the discussion, which I assume stands for conclusion should focus more on the HBV. Other than that I find the article acceptable for publication.

Responses 1: We thank you for your comment and have removed lines 242 - 252, including the removal of Figure 6. We have also changed Figure 7 in the text to Figure 6 (lines 259 and 265). We have also edited the conclusion from line 391 to focus more on HBV. Furthermore, the study participants are newly diagnosed and have not been previously sequenced. We have thus included "were newly diagnosed and" at line 402.